



# Observing carbon dioxide emissions over China's cities with the Orbiting Carbon Observatory-2

Bo Zheng[1], Frederic Chevallier[1], Philippe Ciais[1], Gregoire Broquet[1], Yilong Wang[1, 2], Jinghui Lian[1], and Yuanhong Zhao[1]

[1]Laboratoire des Sciences du Climat et de l'Environnement, CEA-CNRS-UVSQ, UMR8212, Gif-sur-Yvette, France
[2]The Key Laboratory of Land Surface Pattern and Simulation, Institute of Geographical Sciences and Natural Resources Research, Chinese Academy of Sciences, Beijing, China

*Correspondence to*: Bo Zheng (bo.zheng@lsce.ipsl.fr)

**Abstract.** In order to track progress towards the global climate targets, the parties that signed the Paris Climate Agreement
will regularly report their anthropogenic carbon dioxide ($CO_2$) emissions based on energy statistics and $CO_2$ emission factors. Independent evaluation of this self-reporting system is a fast-growing research topic. Here, we study the value of satellite observations of the column $CO_2$ concentrations to estimate $CO_2$ anthropogenic emissions with five years of the Orbiting Carbon Observatory-2 (OCO-2) retrievals over and around China. With the detailed information of emission source locations and the local wind, we successfully observe $CO_2$ plumes from 60 cities and industrial regions over China and
quantify their $CO_2$ emissions from the OCO-2 observations, which add up to a total of 1.6 Gt $CO_2$ yr$^{-1}$ that account for 17% of mainland China's annual emissions. The number of cities whose emissions are constrained by OCO-2 here is three to ten times larger than previous studies that only focused on large cities and power plants in different locations around the world. Our satellite-based emission estimates are broadly consistent with the independent values from the detailed China's emission inventory MEIC, but are more different from those of two widely used global gridded emission datasets (i.e., EDGAR and
ODIAC), especially for the emission estimates for the individual cities. These results demonstrate some skill in the satellite-based emission quantification for isolated source clusters with the OCO-2, despite the sparse sampling of this instrument not designed for this purpose. This skill can be improved by future satellite missions that will have a denser spatial sampling of surface emitting areas, which will come soon in the early 2020s.

## 1 Introduction

The Paris Agreement on climate change requires all parties (countries) to report their anthropogenic greenhouse gas emissions and removals at least every two years within an enhanced transparency framework (UNFCCC, 2018). Then, starting in 2023, the country reports will periodically form the basis for a global stocktake that will assess collective progress in bringing the global greenhouse gas emissions consistent with global warming well below 2°C above pre-industrial levels. In order to address potential biases in this self-reporting mechanism, the contribution of independent observation systems is





being increasingly sought (IPCC, 2019). Our focus here is on the direct observation of fossil fuel carbon dioxide ($CO_2$) emission plumes from space and on the quantification of $CO_2$ emissions from this observation independently.

NASA's second Orbiting Carbon Observatory (OCO-2) polar satellite (Eldering et al., 2017) is one of the best existing instruments for the retrieval of column-averaged dry-air mole fraction of $CO_2$ ($XCO_2$). It observes the clear-sky and sun-lit part of the Earth with the footprints of a few $km^2$ (1.29 km × 2.25 km) gathered in a ~10 km wide swath for each orbit,

particularly suitable for informing natural $CO_2$ budget at the continental scales. It has already acquired more than five years of science data since its launch in July 2014, which has allowed the initial insight into carbon fluxes from the tropical terrestrial ecosystems (Liu et al., 2017; Palmer et al., 2019) but not without ambivalence due to likely significant residual systematic errors in the OCO-2 $XCO_2$ retrievals (Chevallier, 2018).

Extending the use of OCO-2 to monitor fossil fuel $CO_2$ emissions is rather challenging because the excess $XCO_2$ generated

by large cities or power plants typically reaches ~ 1% at best, which is about 4 ppm compared with an instrument noise typically around 1 ppm for a single sounding. This non-negligible noise in the $XCO_2$ retrievals is hardly balanced by the amount of data sampled near emission sources with a narrow swath, which hampers the detection of emission plumes and the precision of emission quantification. Only under rare occasions, the OCO-2 tracks cross $CO_2$ plumes downwind large cities (Labzovskii et al., 2019; Reuter et al., 2019) or power plants (Schwandner et al., 2017; Nassar et al., 2017; Zheng et al.,

2019), limiting the possibility to quantify the corresponding $CO_2$ emissions to few cases within a year. So far, studies on the potential of spaceborne $CO_2$ observation to infer $CO_2$ emissions from large cities or power plants have relied on the Observing System Simulation Experiments (OSSE) (Bovensmann et al., 2010; O'Brien et al., 2016; Broquet et al., 2018; Kuhlmann et al., 2019; Wang et al., 2020) and on several well-chosen cases with real OCO-2 retrievals (Nassar et al., 2017; Reuter et al., 2019; Zheng et al., 2019; Wu et al., 2020). To our knowledge, no attempt has been made yet to infer

anthropogenic emissions from actual satellite data over a large area or a long period to evaluate a large-scale $CO_2$ budget.

Here we analyze all OCO-2 ground tracks between September 2014 and August 2019 over and around China, which is the largest emitter country in the world, in order to quantify $CO_2$ anthropogenic emissions at a large spatial extent over China. We develop a novel, simple, and effective approach to identify the $CO_2$ plumes from isolated emission clusters, to relate them unambiguously to nearby human emission sources, and to estimate the $CO_2$ emission fluxes causing each plume. The

five-year period allows nearly one-fifth of all the emissions from mainland China to be observed, although OCO-2 swaths have a low probability to cross the emission plume from a given city. The budget of $CO_2$ emissions aggregating all the sources inferred from the satellite is compared to different emission inventories compiled by multiplying fuel consumption statistics by emission factors. Such a comparison, for the first time covering a significant fraction of the emissions from a country, demonstrates the potential of independently evaluating the self-reporting emission inventories from space.





## 2 Data and Method

### 2.1 Data input

We use version 9r of the OCO-2 bias-corrected $XCO_2$ retrievals (Kiel et al., 2019). We use the good quality data (xco2_quality_flag equals 0) over both land and ocean, and associated retrieval uncertainty statistics. Our inversion framework relies on the information about the wind and about the spatial distribution of emission sources, which are jointly

used to link the observed $CO_2$ plume section with upwind local emission sources. We choose the spatially explicit Multi-resolution Emission Inventory for China (MEIC) dataset (Zheng et al., 2018a, 2018b) that provides the location of ~100,000 individual industrial point sources (82% of mainland China emissions) and 0.1°×0.1° area source emissions (18% of mainland China emissions) developed for the year 2013. Unlike other inventories used to map industrial emissions using spatial proxies, MEIC includes local reports from each power plant and industrial operator about their emissions and

geographic locations. The ERA5 reanalysis data (C3S, 2017) provides us with a first guess for the local wind fields.

### 2.2 OCO-2 $XCO_2$ local enhancement

The key steps of our method are the identification of an $XCO_2$ local enhancement from the satellite data that can be attributed to a $CO_2$ plume from a large emission source, its separation from the surrounding background, and the establishment of a numerical link to the nearby upwind human emission sources. They are designed to account for the

specificity of the satellite sampling of OCO-2 capability and for the $XCO_2$ retrieval errors.

First, we look for $XCO_2$ anomalies along the OCO-2 tracks, which exceed 2 sigmas of the spatial variability above the local average within 200-km wide moving windows centered at the locations of the anomalies. These anomalies potentially belong to significant $CO_2$ plumes. In each window corresponding to such an anomaly and with more than 200 high-quality retrievals (with ~800 retrievals if none are missing due to cloud contaminations or other issues in the retrieval algorithm), the

following curve fitting is applied to the $XCO_2$ retrieval data along the OCO-2 track:

$$y = m \cdot x + b + \frac{A}{\sigma\sqrt{2\pi}} e^{\left[ -(x-\mu)^2 \big/ 2\sigma^2 \right]} \tag{1}$$

where $y$ is $XCO_2$ (ppm), $x$ is the distance (km) along the OCO-2 track in a fitting window, $m$, $b$, $A$, $\mu$, and $\sigma$ are parameters that determine the curve shape, estimated by a nonlinear least-squares fit weighted by the reciprocal of $XCO_2$ uncertainty statistics. The linear part $m \cdot x + b$ represents the background level assuming the background is linear (Reuter et al., 2019),

while the remaining part depicts a single $XCO_2$ peak with a Gaussian shape (Nassar et al., 2017). Several $XCO_2$ anomalies should belong to the same $CO_2$ plume: in order to only define a single equation for a given plume and the corresponding background, we fit the curve around each $XCO_2$ anomaly and select the one with the largest $R^2$. We also reject all cases with low $R^2$ (less than 0.25) to achieve better fitting performance.

Second, we select the cases when the range of $\mu \pm 3\sigma$ is fully covered by the 200-km window to achieve complete fitting

curves that cover both the plume part and the wide range of local background. To make the curve fitting robust, we further





select the observational cases that have at least 3 valid cross-track footprints (8 footprints if no missing) on average within the plume transect ($\mu \pm 2\sigma$) to constrain the shape of the fitted curve with enough data points. Finally, we check if the parameter $A$ is positive and if the average $XCO_2$ value within $CO_2$ plumes (defined as the average of raw $XCO_2$ retrievals within $\mu \pm 2\sigma$) minus the surrounding background concentration (derived as the average of raw $XCO_2$ retrievals outside $2\sigma$)

is larger than the standard deviation of the background values within 200 km. Only the cases that pass these two filtering criteria are finally identified as the $XCO_2$ local enhancements in this study.

### 2.3 Gaussian plume model

We use the Gaussian plume model (Bovensmann et al., 2010) to attribute the observed $XCO_2$ enhancement to a neighbor cluster of emission sources. We simulate the sum of $XCO_2$ plumes generated by each point source and each emission grid

cell from the MEIC inventory within 50 km of the studied OCO-2 track with equations:

$$V = \sum_{\in 50km} \frac{F}{\sqrt{2\pi} \cdot a \cdot z^{0.894} \cdot u} e^{-\frac{1}{2}\left(\frac{n}{a \cdot z^{0.894}}\right)^2} \qquad (2)$$

$$XCO_2 = V \cdot \frac{M_{air}}{M_{CO2}} \cdot \frac{g}{P_{surf} - w \cdot g} \cdot 1000 \qquad (3)$$

where $V$ is the $CO_2$ vertical column (g m$^{-2}$) downwind of the emission sources, $F$ is the emission rate (g s$^{-1}$), $u$ is the wind speed (m s$^{-1}$), $z$ is the along-wind distance (km), $n$ is the across-wind distance (m), and $a$ is the atmospheric stability

parameter. Equation (3) converts $V$ (g m$^{-2}$) to $XCO_2$ (ppm), where $M$ is the molecular weight (kg mol$^{-1}$), $g$ is the gravitational acceleration (m s$^{-2}$), $P_{surf}$ is the surface pressure (Pa), and $w$ is the total column water vapor (kg m$^{-2}$).

$F$ is derived from the MEIC emission inventory (Zheng et al., 2018b), including both point sources and 0.1°×0.1° area source emissions. Each grid cell of area sources is used as a point source in Equation (2). $u$ is the average wind below 500 m (Beirle et al., 2011) at the time of the OCO-2 overpass, derived from the ERA5 reanalysis data (C3S, 2017). $a$ is a function

of the atmospheric stability condition (Martin, 1976) determined by both the 10-m wind speed and the incoming solar radiation (Seinfeld and Pandis, 2006). Wind, solar radiation, and $P_{surf}$ are all derived from the ERA5 reanalysis dataset (C3S, 2017), and $w$ is adopted from the OCO-2 files.

### 2.4 Cross-sectional $CO_2$ flux estimate

We relate each satellite observed $XCO_2$ enhancement to anthropogenic emission sources within 50 km using the Gaussian

plume model. We visually inspect the observed and modeled $XCO_2$ and further select the ones that exhibit a single and isolated $CO_2$ plume to attribute the plume to a neighbor cluster of emission sources and estimate the corresponding cross-sectional $CO_2$ fluxes. We remove the linear background from the fitted curve of Equation (1) and calculate the area under the remaining fitted curve to derive the $CO_2$ line density (ppm m), which can be converted to the unit of g m$^{-1}$ through Equation (3). The errors in the $CO_2$ line densities are those of the area under the fitted curve, mainly driven by the random errors of the





XCO$_2$ retrievals and also by the Equation (1) that is not a perfect representation of actual CO$_2$ plumes. The standard error statistics for each parameter in Equation (1) are obtained from the weighted nonlinear least-squares fitting, which are propagated to calculate the uncertainties of the area under the fitted curve.

    The CO$_2$ line densities are multiplied by the wind speed (m s$^{-1}$) in the direction normal to the OCO-2 tracks at the location of the plume peak to estimate cross-sectional CO$_2$ fluxes (g s$^{-1}$). The average wind below 500 m is used like in Equation (2). To

reduce the errors in the wind direction, we allow rotation of the wind direction within 45° on each side of the ERA5 local wind direction to maximize the spatial correlation between the Gaussian plume-modeled and the OCO-2-observed XCO$_2$ according to Nassar et al. (2017). The derived cross-sectional CO$_2$ fluxes approximately represent upwind source emissions under steady-state atmospheric conditions, while changes in the atmospheric stability (e.g., strong turbulent diffusion) could make the cross-sectional flux diverge from the source emissions (Varon et al., 2018; Reuter et al., 2019).

**3 Results**

**3.1 CO₂ emission plumes seen by satellite**

    The identification of CO$_2$ emission plumes crossed by the satellite field of view starts with the search for XCO$_2$ local enhancements. These are defined as XCO$_2$ peaks above the background along the thin OCO-2 tracks. As shown in Fig. 1, we have identified a total of 6,565 OCO-2 ground tracks over or around China between September 2014 and August 2019, with

an even share between the cold-season (from September to February, 47%) and the warm-season ones (from March to August, 53%). We find 49,322 XCO$_2$ local enhancements that exceed 2 sigmas above the local average in a 200 km-wide moving window along the satellite tracks. However, 97% of these XCO$_2$ enhancements are removed after evaluation of the integrity of the plume section and of the spatial variation of surrounding background retrievals, leaving only 1,439 XCO$_2$ cases as potent candidates for retrieving emissions.

The second step consists in attempting to attribute the observed 1,439 CO$_2$ enhancements to nearby human emission sources. Only 355 of the 1,439 XCO$_2$ local enhancements can be related to emission sources in the MEIC dataset using the Gaussian plume model within a 50-km upwind distance from each OCO-2 ground track. The other cases that reveal XCO$_2$ enhancement but no nearby emission sources within 50 km upwind are probably due to either OCO-2 XCO$_2$ retrieval errors at local scales, or sources missing in MEIC, or synoptic transport of CO$_2$, over a much longer distance (Parazoo et al., 2011).

The third step is the quantification of cross-sectional CO$_2$ fluxes within the satellite observed CO$_2$ plumes. Only 64 of the 355 cases correspond to single isolated CO$_2$ plumes within a 200 km-wide window, which allow unambiguous attribution to an emission site or cluster. One reason why we reject the other 291 cases is that they have two or more individual plumes (partially overlapping or separated), which are distant in space to make it difficult to merge into a single isolated emission plume transect. Some of the rejected cases also lack observation data of good quality (*xco2_quality_flag* equals 0) at a

distance of several tens of kilometers due to significant retrieval errors in the local satellite observations.



The data filtering process retains more cold-season observations (69%) than warm-season ones, in particular after the last step (56% cases are from the cold season after the second step), due to favorable meteorological patterns during the cold season. The total number of selected cases is several times larger than in previous studies that only focused on large cities and large power plants in different locations of the world (Nassar et al., 2017; Reuter et al., 2019; Wu et al., 2020). The

finally selected 64 cases include both densely populated urban areas and small industrial areas that gather many industrial plants. The peak height of $XCO_2$ enhancement in the plumes ($A/(\sigma\sqrt{2\pi})$ in Equation (1)) is within 0.8–6.0 μmol mol$^{-1}$ (abbreviated as ppm) above the average local background and 2–7 times higher than the standard deviation of background levels within 200 km. The width of observed $CO_2$ plumes, defined as the full width at half maximum of peak height, is estimated between 4.4 and 74.7 km.

**3.2 Quantifying CO₂ emissions: one city example**

Figure 2 presents one example of the 64 selected cases. The emitter here is the city Qinhuangdao that has about one million inhabitants. On October 7$^{th}$ 2018, $CO_2$ emissions from Qinhuangdao were blown southward by a 1.6 m s$^{-1}$ wind at the OCO-2 overpass time and generated an $XCO_2$ local enhancement offshore (Fig. 2). At about local 13:30, OCO-2 flew over the sea to the east of China (Fig. 2a), crossed the $CO_2$ plume transported from Qinhuangdao, and successfully observed the local

enhancement near the northernmost part of OCO-2 ground track (Fig. 2b).

We plot the $XCO_2$ retrieval data (grey dots in Fig. 2c) along the satellite ground track, the plot window of which is centered at the highest $XCO_2$ value in the $CO_2$ plume. We first fit the black curve ($R^2 = 0.7$) based on Equation (1) to depict the $CO_2$ plume transect. The local background is represented by a straight line $-3.3E-5 \cdot x + 404.8$ that approximates a flat background of 404.8 ppm. Then we subtract the background line from both the $XCO_2$ data and the fitted black curve to

obtain the net enhancement of $XCO_2$ above the local background (pink dots and red curve in Fig. 2d). The maximum $XCO_2$ net enhancement (peak height of the red curve) is 2.7 ppm and the plume width is 37.4 km. The $CO_2$ line density is estimated as $1.7 \pm 0.1$ t-$CO_2$ m$^{-1}$ (central estimate ± 1σ) by computing the area under the red curve (the orange shade in Fig. 2d). The uncertainty is mainly caused by random errors of the single $XCO_2$ retrievals.

The $CO_2$ line density derived from the satellite retrievals is further multiplied by the wind speed in the normal direction to

the OCO-2 track to quantify the cross-sectional $CO_2$ fluxes. We use the average wind below 500 m from the ERA5 reanalysis data. The ceiling height of 500 m is comparable to the maximum height that smoke plumes from power plants and industrial plants typically reach. The wind direction around Qinhuangdao is optimized according to Nassar et al. (2017) and is shifted by 1° to maximize the spatial correlation between the satellite-observed (Fig. 2d) and the model-simulated (Fig. 2e) $XCO_2$ enhancements. The wind speed in the normal direction to the OCO-2 track is then estimated as 0.6 m s$^{-1}$ at the

location of the maximum $XCO_2$ value (Fig. 2b). The $CO_2$ hourly flux at the satellite overpassing time is finally estimated as $3.4 \pm 0.7$ kt-$CO_2$ h$^{-1}$, considering uncertainties both in the $CO_2$ line density and in the wind speed.

The satellite observed $CO_2$ plume can be traced back to anthropogenic emission sources located in the urbanized area of the Qinhuangdao city by the Gaussian plume model combined with the local emission map given by the MEIC inventory. We





use monthly, weekly, and diurnal emission time profiles by region and by source sector from MEIC to split the annual
emission totals reported by MEIC to hourly emission rates during the satellite overpass. The MEIC hourly emission rate of
Qinhuangdao is $2.6 \pm 0.8$ kt-$CO_2$ h$^{-1}$, which is close to the satellite-based inversion estimate.

### 3.3 $CO_2$ emission estimates for 60 regions in China

We quantify the $CO_2$ emissions corresponding to the 64 $CO_2$ plumes selected from the five-year OCO-2 archive. These
represent 60 different urban areas or industrial regions in China. There are 4 regions whose emission plumes were observed
twice in our selection of the satellite data. The 64 $CO_2$ plumes present $CO_2$ line densities between 0.1 and 2.8 t-$CO_2$ m$^{-1}$, and
hourly $CO_2$ fluxes at the time of the satellite overpass are estimated within the range of 0.2–15.4 kt-$CO_2$ h$^{-1}$ with the $1\sigma$
uncertainties of 20–28%. The larger sources tend to present lower relative uncertainties, because a larger X$CO_2$ enhancement
makes it easier to separate a plume from its background, and is thus more easily observed by the satellite. The inversions that
estimate $CO_2$ emissions larger than 4 kt-$CO_2$ h$^{-1}$ tend to constrain their relative uncertainties below 24%.
We compare the satellite-based $CO_2$ hourly fluxes to the corresponding source emissions given by MEIC (Fig. 3), after
applying emission time profiles to transform MEIC annual emissions into hourly emissions at the time of satellite overpass.
Although the point source based MEIC emissions data is only for the year 2013, China's countrywide emissions remained
stable between 2013 and 2017 and marginally grew only after 2017 (Friedlingstein et al., 2019). The satellite-based and
MEIC estimated emissions are broadly consistent within a factor of two (solid dots in Fig. 3) with comparable uncertainties
for the same individual estimates. Both approaches estimate the same average $CO_2$ flux of the 64 emission plumes as 3.8 kt-
$CO_2$ h$^{-1}$. The average of satellite-based estimates is 5.5% higher than the MEIC values in the cold season (solid blue dots in
Fig. 3), while 6.0% lower in the warm season (solid red dots in Fig. 3).
The satellite-based larger estimates in the cold season are partially due to the fact that human respiration contributes to urban
$CO_2$ fluxes while not included in the MEIC inventory of fossil fuel and cement emissions. We make a rough estimate of the
metabolic $CO_2$ release by multiplying an emission factor of 0.52 t-$CO_2$ yr$^{-1}$ person$^{-1}$ (Prairie and Duarte, 2007) by the
population living in each emitting area. The results suggest that human metabolic $CO_2$ emissions explain 38% of the larger
satellite-based emission estimates on average in the cold season. The remaining difference could be due to natural processes
like plant respiration or to the slight growth of fossil fuel emissions since 2013, but could also reflect some bias in the MEIC
estimates. In the warm season, despite human respiration emissions, the satellite-based inversions give lower emission
estimates possibly due to the carbon uptake by urban green spaces that are not separated from anthropogenic emissions in the
satellite-based inversion method.
The uncertainties in the satellite-based emission estimates are driven by those of the local wind field and of the $CO_2$ line
density derived from the X$CO_2$ retrievals. We reduce the errors in wind directions and consequently increase the $R^2$ of the
linear correlation between satellite- and MEIC-based emission estimates across emitting areas from 0.16 (open dots) to 0.57
(solid dots) as shown in Fig. 3. The magnitude of the wind speed uncertainty, typically considered 10–20% (Nassar et al.,
2017; Varon et al., 2018; Reuter et al., 2019), is comparable to the uncertainty in the satellite-based $CO_2$ line densities (4–19%



for the 64 emission plumes). In high wind-speed conditions, the $CO_2$ plumes are spread more quickly and thus cause smaller local enhancements, which weakens the signal of $XCO_2$ and causes larger uncertainties in the estimate of $CO_2$ line densities. Generally, our estimates reach lower relative uncertainties for larger emission cities under lower wind speeds.

### 220  3.4 Comparison with global bottom-up inventories

We extrapolate the satellite-based $CO_2$ hourly fluxes to annual total fluxes using emission time profiles, and compare them to two global bottom-up emission maps: ODIAC (Oda and Maksyutov, 2015) and EDGAR (Janssens-Maenhout et al., 2019). We use the cases between the years 2014 and 2018 when both inventories are available, and extract $CO_2$ emissions over each satellite-observed emitting area from the emission maps (Fig. 4). For the areas observed by the satellite in different years, we
compute annual values from the corresponding inversions and average them for the comparison with ODIAC and EDGAR. For individual estimates, ODIAC (Figs. 4b) and EDGAR (Figs. 4c) are broadly consistent with the annual budgets from the satellite-based inversions, but the fit is slightly better in the case of EDGAR. For Beijing where MEIC remarkably agrees with the satellite estimate (Fig. 3), ODIAC and EDGAR estimate larger emissions by 322% and 202%, respectively. Such large discrepancies are not surprising since global emission inventories typically involve large uncertainties at city scales
(Gately and Hutyra, 2017; Gurney et al., 2019), because they disaggregate national emissions to gridded maps with simple proxies like population or nighttime light in the countries like China where they lack detailed direct local information. Only large power plants have exact geographic locations (from the CARMA global database (Wheeler and Ummel, 2008)), in principle, not all of the industrial plants like MEIC. Such an emission mapping approach overestimates emissions over densely populated cities in China (Zheng et al., 2017), because the industry plants, the primary $CO_2$ emission sources in
China, are located far away from densely urban areas.
The sum of the emissions from the satellite-observed areas reaches 1.58 Gt $CO_2$ yr$^{-1}$ (Fig. 4a), accounting for 17% of the mainland China's total emissions. The corresponding bottom-up estimates from ODIAC, EDGAR, and MEIC are 1.39, 1.55, and 1.52 Gt $CO_2$ yr$^{-1}$, respectively. ODIAC emissions are 12% lower than the satellite-based estimates while EDGAR emissions are only 2% lower. The slight growth of the emissions from 2014 to 2018 (documented in, e.g., EDGAR) could
alone explain the 4% lower value for MEIC (valid for the year 2013) than the satellite estimate. Overall, EDGAR matches the estimate from the satellite-based inversions better than ODIAC for the 17% of mainland China's $CO_2$ emissions that are observed by the satellite. However, both of these two global emission inventories reveal large uncertainties in emission estimates for individual areas as shown in Figs. 4b and 4c.

### 4. Discussion

We developed a novel objective approach to quantify local anthropogenic $CO_2$ emissions from the OCO-2 $XCO_2$ satellite retrievals. The key of this method is a conservative selection of the satellite data that can be safely exploited for emission quantification. It also depends on the wind information and the information about the locations of human emission sources in



the upwind vicinity of the selected OCO-2 tracks. Future developments could aim at refining the stringent data selection, or at improving the description of the plume footprint, for instance using detailed regional atmospheric transport models but the

current simplicity of our approach makes it easily applicable everywhere over the globe in principle. Our first regional analysis over mainland China suggests that 17% of its $CO_2$ human emissions can be observed and constrained, to some extent, by five years of retrieval data from the OCO-2, a satellite instrument not designed for this task. The satellite-based emission inversion results are broadly consistent ($R^2$=0.57, meaning we agree on broad classes of emitters) with the reliable point source-based MEIC regional inventory despite our simple modeling of the plume and of its background, and despite

possible biases due to local non-fossil fuel emissions or local sinks that contribute to the plume intensity. We also use the satellite-based estimates as a rough independent evaluation of two global bottom-up inventories, ODIAC and EDGAR.

There is still a large gap between what the satellite can see and what the National Greenhouse Gas Inventory reports submitted to the United Nations Framework Convention on Climate Change (UNFCCC), mentioned at the start of the introduction. The former is made of specific emission plumes linked to recent emission events without any sectoral

distinction within the plume. The latter is made of the country- and annual-scale emission values assigned to specific human-caused source/sink categories. The exhaustiveness of the MEIC inventory, which involved detailed analysis of the fine spatial and temporal emission patterns, allowed us to bridge most of this gap for a time period when Chinese emissions did not vary much, but few countries have such a detailed geospatial inventory of their emissions and are able to update it timely for such a task. We also acknowledge the limitations of the emission temporal profiles even from the detailed MEIC

inventory. The sparse sampling of the OCO-2 instrument, despite the good precision of individual soundings, will partly be overcome by the next-generation of $CO_2$-dedicated imagery satellites, such as the $CO_2$ Monitoring mission (CO2M) in Europe (Clery, 2019) and the Geostationary Carbon Cycle Observatory (GeoCarb) in the U.S. (Moore III et al., 2018) that will have denser spatial coverage. However, their measurement principle still relies on sunlight and will prevent us from well sampling the emission diurnal cycle. The need for a good knowledge of the emission space-time patterns (not only the

emission values) will therefore remain for the comparison between the national inventories and the satellite-based estimates. However, for countries with less $CO_2$ inventory infrastructures (typically non-Annex I parties to UNFCCC), we could also envisage an incremental approach where both bottom-up and top-down estimates are developed together in parallel.

**Data availability**

The version 9r of the OCO-2 bias-corrected $XCO_2$ retrievals were downloaded from the data archive maintained at the

NASA Goddard Earth Science Data and Information Services Center (https://oco2.gesdisc.eosdis.nasa.gov/data/s4pa/OCO2_DATA/OCO2_L2_Lite_FP.9r/, last access: 12 February 2020; Kiel et al., 2019). The ERA5 reanalysis data were acquired from the Copernicus Climate Change Service Climate Data Store (https://cds.climate.copernicus.eu/, last access: 12 February 2020; C3S, 2017).





**Author contributions**

BZ, FC, and PC designed the study. BZ processed the observational data and estimated the $CO_2$ fluxes from the satellite observations. BZ prepared the paper with contributions from all coauthors.

**Competing interests**

The authors declare that they have no conflict of interest.

**Acknowledgements**

The OCO-2 retrievals were produced by the OCO-2 project at the Jet Propulsion Laboratory, California Institute of Technology, and obtained from the OCO-2 data archive maintained at the NASA Goddard Earth Science Data and Information Services Center.

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




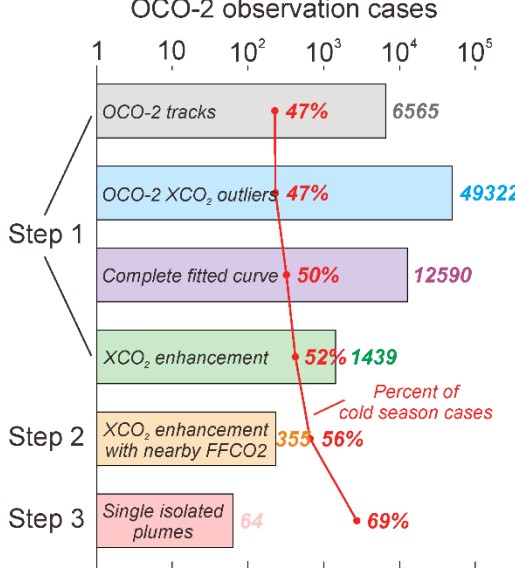


**Figure 1: OCO-2 XCO₂ observation cases contained in each processing step.** Step 1 starts from 6,565 OCO-2 tracks around and over China between September 2014 and August 2019 (grey bar) and finds 49,322 XCO₂ anomalies along the OCO-2 tracks (blue bar). 12,590 anomalies (purple bar) and their surrounding data points within a 200 km-wide window can be fitted by a complete nonlinear curve using Equation (1), of which 1,439 XCO₂ anomalies (green bar) are identified as local enhancement significantly higher than the background. Step 2 uses the Gaussian plume model to select 355 XCO₂ enhancements (yellow bar) that can be traced back to upwind fossil fuel emission sources within 50 km. In step 3, we finally select the 64 cases with single isolated CO₂ plumes to quantify the CO₂ emissions. The red curve shows the percentage of cold-season observational cases in each bar. The detail of each step is described in Sect. 2.



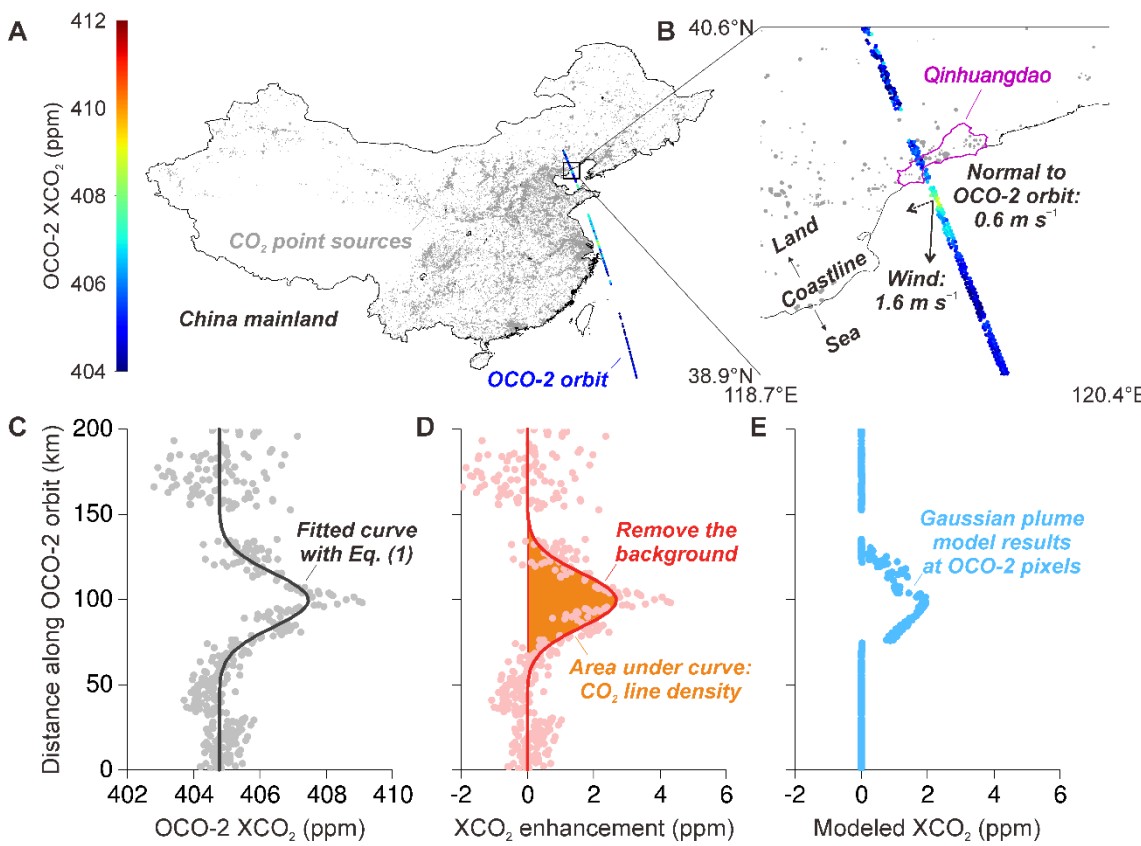

**Figure 2: Quantification of CO₂ emissions from Qinhuangdao.** (A) The OCO-2 orbit on October 7th 2018 is plotted on the map of
MEIC emission point sources. (B) Zoom in closer to see OCO-2 XCO₂ data, local wind speed, and wind direction. The width of the track
is made of eight cross-track OCO-2 footprints. (C) The valid XCO₂ data points (grey dots) plotted along the OCO-2 orbit with a fitted
curve (black) based on Equation (1). (D) The XCO₂ enhancement (red dots) above background, the fitted curve (red), and the area under
the curve (orange shade). (E) The modeled XCO₂ enhancement (blue dots) by the Gaussian plume model combined with the MEIC
emission inventory.





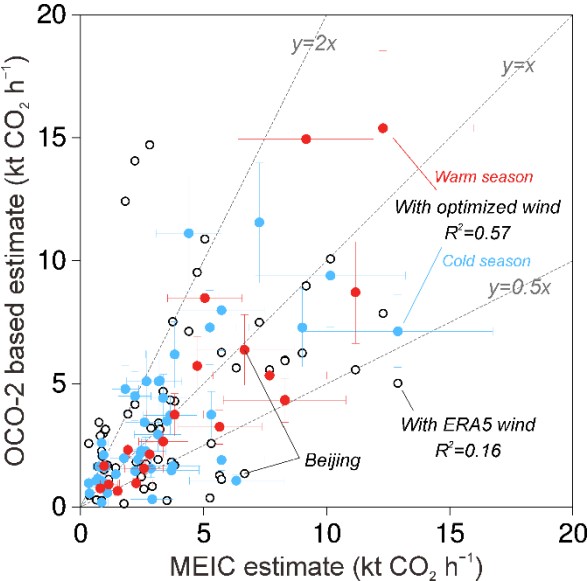


**Figure 3: Comparison between OCO-2 based and MEIC estimated $CO_2$ hourly fluxes.** Each dot represents one of the 64 plume cases selected in this study, plotted according to the MEIC estimated $CO_2$ flux (x-axis) and the OCO-2-based estimate (y-axis). The open dots are OCO-2 estimates using the ERA5 wind data, while the solid dots use the optimized wind and distinguish the warm-season (red dots) and the cold-season (blue dots) cases.





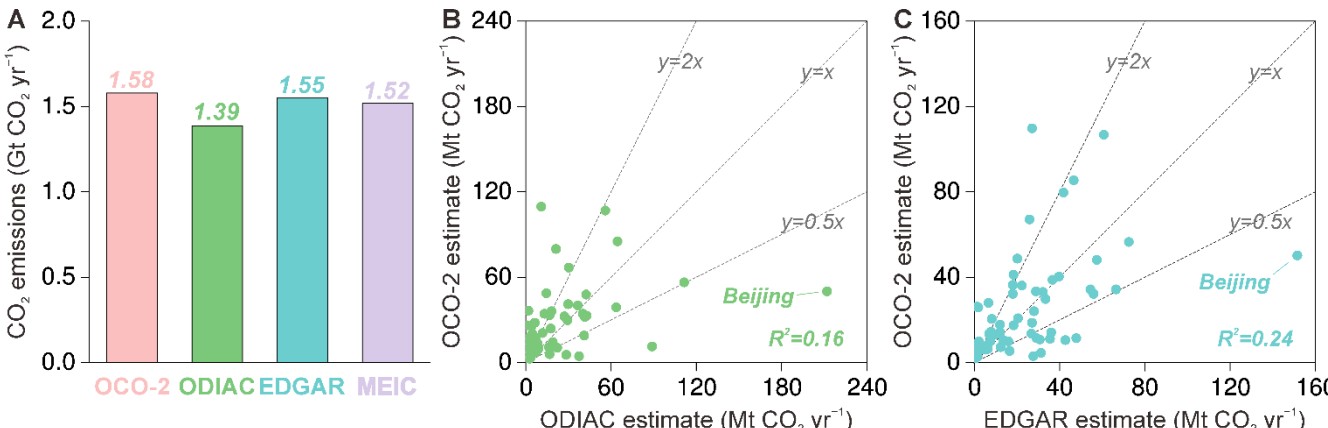


**Figure 4: Comparing OCO-2 based CO₂ emission estimates with bottom-up inventories.** (A) The sum of emissions from the 60 regions observed by OCO-2 between the years 2014 and 2018, including OCO-2 estimates (scaled up to annual emissions based on MEIC emission time profiles, pink bar), ODIAC (green bar), EDGAR (blue bar), and MEIC (purple bar) estimates. (B) Comparison of regional CO₂ emissions between OCO-2-based (y-axis) and ODIAC estimates (x-axis). (C) Comparison of regional CO₂ emissions between OCO-415 2-based (y-axis) and EDGAR estimates (x-axis).