# Peer review of "Observing carbon dioxide emissions over China's cities and industrial areas with the Orbiting Carbon Observatory-2"

_Atmospheric Chemistry and Physics, 2020_

## Referee Comment (RC1) · Anonymous Referee #1 · 13 Apr 2020

General comments.

The manuscript reports good progress in quantifying multiple megacity emissions of CO2 in China using a plume transport model and CO2 observations by OCO-2 satellite. The mean estimate of the emissions from selected megacity areas is comparable with inventory data. The manuscript is well written and can be recommended for publication after minor revisions, taking into the account the following comments:

Detailed comments.

Line 42 As for instrument noise (not retrieval noise) it may be better to use a number in the order of 0.3 - 0.6 ppm as in (Worden et al., 2017)

Line 49 Authors write "To our knowledge, no attempt has been made yet to infer anthropogenic emissions from actual satellite data over a large area or a long period to evaluate a large-scale CO2 budget." Suggest being more specific here and write as "actual OCO-2 data", otherwise, when speaking about satellites, there is a study by Janardanan et al., (2016) using several years of CO2 data for assessing emissions from large regions. Also adding somewhere reference to Kort et al., 2012 is useful from historical context.

Line 176 "The ceiling height of 500 m is comparable to the maximum height that smoke plumes from power plants and industrial plants typically reach." The assumption seems to be weak, as turbulent mixing is supposed to mix CO2 up to PBL top, exceeding 500 m in many occasions. The practical choice of using a mean wind vector below 500 m may be driven by other reasons.

Line 222 More informative reference to ODIAC is given by Oda et al., (2018)

Line 267 For CO2-M there is a recent mission paper by Janssens-Maenhout et al. (2020)

Line 210 Summertime uptake by green spaces in a city should not be used as an explanation here as vegetation uptake is also present in the background used as reference for estimating enhancements.

Line 235 There is an impression that there is a 200-300% disagreement between MEIC and other inventories in cities, and it is caused by misplacing industrial emissions. There are other factors apart from placing industrial emissions. ODIAC is using a simple disaggregation of emissions by using nightlights, which may lead to underestimation of road emissions, as found by Gateley and Hutyra (2017), so it is supposed to be missing some emissions in cites still it was found by Gateley and Hutyra (2017) to correlate well with the detailed inventory at 5 km resolution. EDGAR inventory is not supposed to suffer from misplacing industrial emissions to the same extent as ODIAC thus there should be another reason for disagreement. A reader would benefit from providing more details on scale and reason for discrepancies between the inventories

in the target areas.

References

Janardanan, R., Maksyutov, S., Oda, T., Saito, M., Kaiser, J. W., Ganshin, A., Stohl, A., Matsunaga, T., Yoshida, Y., and Yokota, T.: Comparing GOSAT observations of localized CO2 enhancements by large emitters with inventory-based estimates, Geophys. Res. Lett., 43, 3486-3493, doi:10.1002/2016GL067843, 2016.

Janssens-Maenhout, G., B. Pinty, M. Dowell, H. Zunker, E. Andersson, et al: Towards an operational anthropogenic CO2 emissions monitoring and verification support capacity. Bull. Amer. Meteor. Soc., https://doi.org/10.1175/BAMS-D-19-0017.1, 2020.

Kort, E. A., Frankenberg, C., Miller, C. E., and Oda, T.: Space-based observations of megacity carbon dioxide, Geophys. Res. Lett., 39, L17806, doi:10.1029/2012GL052738., 2012.

Oda, T., Maksyutov, S., and Andres, R. J.: The Open-source Data Inventory for Anthropogenic CO2, version 2016 (ODIAC2016): a global monthly fossil fuel CO2 gridded emissions data product for tracer transport simulations and surface flux inversions, Earth Syst. Sci. Data, 10, 87–107, https://doi.org/10.5194/essd-10-87-2018, 2018.

Worden, J. R., Doran, G., Kulawik, S., Eldering, A., Crisp, D., Frankenberg, C., O'Dell, C., and Bowman, K.: Evaluation and attribution of OCO-2 XCO2 uncertainties, Atmos. Meas. Tech., 10, 2759–2771, https://doi.org/10.5194/amt-10-2759-2017, 2017.

---

## Referee Comment (RC2) · Anonymous Referee #2 · 20 Apr 2020

The study by Zheng et al. uses the complete XCO2 data record available from the OCO-2 satellite instrument to estimate the CO2 emissions of 60 individual sources (cities, power plants, industrial areas) in China, accounting for almost one fifth of China's total CO2 emissions. Several previous studies showed the potential of OCO-2 to detect and quantify strong point sources, but those studies were demonstrations rather than systematic analyses of OCO-2's ability to quantify regional emissions as presented here. The study by Zheng et al. is thus an important step forward. The applied methods are thorough and convincing. I particularly appreciated the conservative and careful selection of cases, for which emission quantification was safely possible. The results of the study nicely demonstrate the potential but also the great challenges offered by spaceborne CO2 observations for emission quantification.

[Figure]

Emissions were estimated in the same way as in previous studies, i.e. by computing the integral amount of $CO_2$ in cross-sections through the plume multiplied by the wind component perpendicular to these cross-sections. However, there are novel elements that go beyond previous studies, notably the combination of a detailed emission inventory for China with a Gaussian plume approach where sources (e.g. cities) are not treated as individual plumes but as superpositions of multiple plumes emanating from individual area and point sources. Although the information from these super-positioned plumes was not used directly for plume quantification, it was used to attribute the plumes to specific emission sources, which was a critical step in the selection of suitable cases.

Overall, the paper is very well written and an important contribution to the growing literature on the quantitative interpretation of OCO-2 observations. I thus support publication after addressing the following points.

Main points:

- The title of the manuscript suggests that the study is about emissions of cities. However, the 60 plumes are not only from cities but also from "industrial regions". The authors should state explicitly how many of these plumes were representing emissions from cities, power plants and industrial complexes. This is important information for the planning of future satellite missions, since it is still not clear how well plumes from cities can be ob-served in comparison to those from power plants.

- The choice of a maximum distance of 50 km (page 4, line 100) between sources and OCO-2 track seems rather arbitrary. How does this choice affect the results? 50 km seems a rather short distance. More distant sources could contribute to the plumes and bias the corresponding estimates. The model-based study of Kuhlmann et al. (https://doi.org/10.5194/amt-12-6695-2019), for example, demonstrated that the plume of a power plant (Jänschwalde) 100 km away from a city (Berlin) could significantly overlap with the city plume in some cases.

- According to Bieser et al. (https://doi.org/10.1016/j.envpol.2011.04.030), roughly 90% of emissions from power plants occur between 200 m and 500 m above surface. How would emission estimates for power plants change using an average wind speed over this range rather than an average over 0 – 500 m (page 6, line 176)? Note that at the small distances between source and OCO-2 track considered in this study one cannot expect a homogeneous mixing of the plume over the depth of the PBL.

Minor points and grammar:

- Page 1, line 18: Change "from the detailed China's emission inventory" to "from China's detailed emission inventory"

- P2, L34: "with the footprints" –> "with footprints"

- P2, L35: "natural $CO_2$ budget" -> "natural $CO_2$ budgets"

- P2, L36: "has allowed the initial insight" -> "has provided initial insight"

- P2, L46: "spaceborne $CO_2$ observation" -> "spaceborne $CO_2$ observations"

- P3, L64: "relies on the information about the wind" -> "relies on auxiliary information about winds"

- P3, L66: "provides the location" -> "provides the locations"

- P3, L75: "satellite sampling of OCO-2 capability" -> "sampling capability of OCO-2"

- P3, L77: "centered at the locations" -> "centered on the locations"

- P3, L86: Why should several $XCO_2$ anomalies belong to the same $CO_2$ plume? There is only a single transect per plume. Because of the moving windows?

- P4, L91: "8 footprints if no is missing" -> "8 footprints if none is missing"

- P4, L93: "within $CO_2$ plume" -> "within the plume"

- P4, L104: Only a detail: Why is the along-wind distance measured in kilometres, but

the across-wind distance in meters?

- P5, L136: I think it would be clearer to state "We find 49,322 cases with local XCO2 enhancements". It wasn't clear to me initially whether these were individual pixels or plumes.

- P5, L144: 50 km is not an appropriate scale for synoptic transport. I suggest to simplify to ".. or transport of CO2 over a longer distance"

- P5, L148: "in space to make it difficult" -> "in space making it difficult"

- P6, L153: It would be better to write "Although the total number of selected cases is small, it is several times larger .."

- P6, L163: "at about local 13:30" -> "at about 13:30 local time"

- P6, L164: "part of OCO-2 ground track" -> "part of the OCO-2 ground track"

- P6, L175: "CO2 fluxes" -> "CO2 flux"

- P6, L178: Why shifted by 1°? Maybe it would be clearer to state "shifted by 1° in this case".

- P7, L186: How was the uncertainty of the hourly emission rate of Qinhuangdao determined? Does the MEIC inventory include uncertainties?

- P7, L194: There were 4 cases where the same source was quantified twice. It would be good to know how consistent those double quantifications are with the estimated uncertainty of <24%.

- P7, L203ff: The interpretation of the small differences of 5-6% between satellite based estimates and MEIC in different seasons is pushed too far in this section considering the uncertainties. At least the arguments should be presented as possible explanations rather than as facts (e.g. write "could be due to" rather than "are due to"). The over-interpretation of the results culminates in the statement that human respiration

accounts for 38% of the (5.5%) difference and that the remaining difference could be due to a bias in MEIC. The numbers deduced from the satellite observations are not sufficiently robust to speculate about a bias in the inventory as small as 3 percent. Uncertainties in the method (notably the assumption that the 0-500 m average wind speed is representative) could easily explain such differences, probably also differences in the results between summer and winter since vertical mixing is different in these seasons.

- P8, L233: "principle, not all" -> "principle, but not all"

- P8, L235: "densely urban areas" -> "densely populated urban areas"

- P8, L44: The last section should be renamed to "Conclusions".

- P9, L271: "with less CO2 inventory infrastructures" -> "with less advanced CO2 inventory infrastructures"

- Figure 1: The blue bar should be called "XCO2 anomalies" rather than "XCO2 outliers"

---

## Author Comment (AC1) · 21 Jun 2020

**Referee #1:**

General comments.

The manuscript reports good progress in quantifying multiple megacity emissions of $CO_2$ in China using a plume transport model and $CO_2$ observations by OCO-2 satellite. The mean estimate of the emissions from selected megacity areas is comparable with inventory data. The manuscript is well written and can be recommended for publication after minor revisions, taking into the account the following comments:

**Response:**

We thank the referee for the positive comments on our manuscript.

Detailed comments.

Line 42 As for instrument noise (not retrieval noise) it may be better to use a number in the order of 0.3 - 0.6 ppm as in (Worden et al., 2017)

**Response:**

This sentence has been rewritten as "an instrument noise typically around 0.3–0.6 ppm (Worden et al., 2017)".

Line 49 Authors write "To our knowledge, no attempt has been made yet to infer anthropogenic emissions from actual satellite data over a large area or a long period to evaluate a large-scale $CO_2$ budget." Suggest being more specific here and write as "actual OCO-2 data", otherwise, when speaking about satellites, there is a study by Janardanan et al., (2016) using several years of $CO_2$ data for assessing emissions from large regions. Also adding somewhere reference to Kort et al., 2012 is useful from historical context.

**Response:**

We now use the expression "actual OCO-2 data" and have added the reference to Kort et al. (2012).

Line 176 "The ceiling height of 500 m is comparable to the maximum height that smoke plumes from power plants and industrial plants typically reach." The assumption seems to be weak, as turbulent mixing is supposed to mix $CO_2$ up to PBL top, exceeding 500 m in many occasions. The practical choice of using a mean wind vector below 500 m may be driven by other reasons.

**Response:**

To quantify cross-sectional $CO_2$ fluxes, we need to know the horizontal wind direction and wind speed at the $CO_2$ plume height (Nassar et al., 2017). For a power plant, Nassar et al. (2017) used the wind vector at the stack height. Since this study focuses on cities that have emission sources with various stack heights, we used the average wind below 500 m following Beirle et al. (2011).

Line 222 More informative reference to ODIAC is given by Oda et al., (2018)

**Response:**

We have added the reference to Oda et al. (2018) in the revised manuscript.

Line 267 For CO2-M there is a recent mission paper by Janssens-Maenhout et al. (2020)

**Response:**

We have added the reference to Janssens-Maenhout et al. (2020).

Line 210 Summertime uptake by green spaces in a city should not be used as an explanation here as vegetation uptake is also present in the background used as reference for estimating enhancements.

**Response:**

The $XCO_2$ enhancement tends to be lower in summer than in winter (Mitchell et al., 2018) due to the photosynthetic uptake by plants. This phenomenon makes $FFCO_2$ signal not easily separated from the surrounding background, which could partly explain the slight underestimates in the $FFCO_2$ fluxes from OCO-2 $XCO_2$ retrievals in summer. We have clarified it in the manuscript.

Line 235 There is an impression that there is a 200-300% disagreement between MEIC and other inventories in cities, and it is caused by misplacing industrial emissions. There are other factors apart from placing industrial emissions. ODIAC is using a simple disaggregation of emissions by using nightlights, which may lead to underestimation of road emissions, as found by Gateley and Hutyra (2017), so it is supposed to be missing some emissions in cites still it was found by Gateley and Hutyra (2017) to correlate well with the detailed inventory at 5 km resolution. EDGAR inventory is not supposed to suffer from misplacing industrial emissions to the same extent as ODIAC thus there should be another reason for disagreement. A reader would benefit from providing more details on scale and reason for discrepancies between the inventories in the target areas.

**Response:**

We provide a brief discussion on the discrepancies between MEIC and other inventories as follows.

"The large discrepancies are not surprising since global emission inventories typically involve large uncertainties at city scales (Gately and Hutyra, 2017; Gurney et al., 2019), because they disaggregate national emissions to gridded maps with simple proxies like population or nighttime light in the countries like China where they lack detailed direct local information. Only large power plants have exact geographic locations (from the CARMA global database (Wheeler and Ummel, 2008)), in principle, not all of the industrial plants like MEIC. The ODIAC uses nightlights to disaggregate national emission estimates to grid cells, which may lead to an underestimation of road emissions in cities (Gately and Hutyra, 2017) and a misplacing of industrial emissions. The EDGAR relies on point source locations to allocate emissions in space while it still suffers from missing local information in China, and gridded population maps have to be used instead. Such an emission mapping approach overestimates emissions over densely populated cities in China (Zheng et al., 2017), because the industry plants, the primary $CO_2$ emission sources in China, are located far away from densely urban areas. The MEIC inventory estimates industrial emissions at the facility scale, transport emissions at the county scale, and residential emissions at the provincial scale, which can achieve better spatial accuracy in emissions estimates than the global emission inventories."

**References**

Beirle, S., Boersma, K. F., Platt, U., Lawrence, M. G., and Wagner, T.: Megacity Emissions and Lifetimes of Nitrogen Oxides Probed from Space, Science, 333, 1737-1739, doi: 10.1126/science.1207824, 2011.

Mitchell, L. E., Lin, J. C., Bowling, D. R., Pataki, D. E., Strong, C., Schauer, A. J., Bares, R., Bush, S. E., Stephens, B. B., Mendoza, D., Mallia, D., Holland, L., Gurney, K. R., and Ehleringer, J. R.: Long-term urban carbon dioxide observations reveal spatial and temporal dynamics related to urban characteristics and growth, Proc. Natl. Acad. Sci., 115, 2912-2917, doi: 10.1073/pnas.1702393115, 2018.

Nassar, R., Hill, T. G., McLinden, C. A., Wunch, D., Jones, D. B. A., and Crisp, D.: Quantifying $CO_2$ Emissions From Individual Power Plants From Space, Geophys. Res. Lett., 44, 10,045-010,053, doi: 10.1002/2017gl074702, 2017.

References

Janardanan, R., Maksyutov, S., Oda, T., Saito, M., Kaiser, J. W., Ganshin, A., Stohl, A., Matsunaga, T., Yoshida, Y., and Yokota, T.: Comparing GOSAT observations of localized $CO_2$ enhancements by large emitters with inventory-based estimates, Geophys. Res. Lett., 43, 3486-3493, doi:10.1002/2016GL067843, 2016.

Janssens-Maenhout, G., B. Pinty, M. Dowell, H. Zunker, E. Andersson, et al: Towards an operational anthropogenic CO2 emissions monitoring and verification support capacity. Bull. Amer. Meteor. Soc., https://doi.org/10.1175/BAMS-D-19-0017.1, 2020.

Kort, E. A., Frankenberg, C., Miller, C. E., and Oda, T.: Space-based observations of megacity carbon dioxide, Geophys. Res. Lett., 39, L17806, doi:10.1029/2012GL052738., 2012.

Oda, T., Maksyutov, S., and Andres, R. J.: The Open-source Data Inventory for Anthropogenic $CO_2$, version 2016 (ODIAC2016): a global monthly fossil fuel $CO_2$ gridded emissions data product for tracer transport simulations and surface flux inversions, Earth Syst. Sci. Data, 10, 87–107, https://doi.org/10.5194/essd-10-87-2018, 2018.

Worden, J. R., Doran, G., Kulawik, S., Eldering, A., Crisp, D., Frankenberg, C., O'Dell, C., and Bowman, K.: Evaluation and attribution of OCO-2 $XCO_2$ uncertainties, Atmos. Meas. Tech., 10, 2759–2771, https://doi.org/10.5194/amt-10-2759-2017, 2017.

---

## Author Comment (AC2) · 21 Jun 2020

**Referee #2:**

The study by Zheng et al. uses the complete $XCO_2$ data record available from the OCO-2 satellite instrument to estimate the $CO_2$ emissions of 60 individual sources (cities, power plants, industrial areas) in China, accounting for almost one fifth of China's total $CO_2$ emissions. Several previous studies showed the potential of OCO-2 to detect and quantify strong point sources, but those studies were demonstrations rather than systematic analyses of OCO-2's ability to quantify regional emissions as presented here. The study by Zheng et al. is thus an important step forward. The applied methods are thorough and convincing. I particularly appreciated the conservative and careful selection of cases, for which emission quantification was safely possible. The results of the study nicely demonstrate the potential but also the great challenges offered by spaceborne $CO_2$ observations for emission quantification.

Emissions were estimated in the same way as in previous studies, i.e. by computing the integral amount of $CO_2$ in cross-sections through the plume multiplied by the wind component perpendicular to these cross-sections. However, there are novel elements that go beyond previous studies, notably the combination of a detailed emission inventory for China with a Gaussian plume approach where sources (e.g. cities) are not treated as individual plumes but as superpositions of multiple plumes emanating from individual area and point sources. Although the information from these super-positioned plumes was not used directly for plume quantification, it was used to attribute the plumes to specific emission sources, which was a critical step in the selection of suitable cases.

Overall, the paper is very well written and an important contribution to the growing literature on the quantitative interpretation of OCO-2 observations. I thus support publication after addressing the following points.

**Response:**

We thank the referee for the constructive and positive comments on our paper. We provide point-by-point responses as follows.

Main points:

- The title of the manuscript suggests that the study is about emissions of cities. However, the 60 plumes are not only from cities but also from "industrial regions". The authors should state explicitly how many of these plumes were representing emissions from cities, power plants and industrial complexes. This is important information for the planning of future satellite missions, since it is still not clear how well plumes from cities can be observed in comparison to those from power plants.

**Response:**

Among the 60 plumes that we analyzed, 33 plumes are from cities and the other 27 ones are from industrial regions. We now clarify this in the Sect. 3.1 as follow.

"The finally selected 60 cases include both densely populated urban areas (33 cases) and small industrial areas (27 cases) that gather many industrial plants."

We also revise the title of our paper to "Observing carbon dioxide emissions over China's cities and industrial areas with the Orbiting Carbon Observatory-2".

- The choice of a maximum distance of 50 km (page 4, line 100) between sources and OCO-2 track seems rather arbitrary. How does this choice affect the results? 50 km seems a rather short distance. More distant sources could contribute to the plumes and bias the corresponding estimates. The model-based study of Kuhlmann et al. (https://doi.org/10.5194/amt-12-6695-2019), for example, demonstrated that the plume of a power plant (Jänschwalde) 100 km away from a city (Berlin) could significantly overlap with the city plume in some cases.

**Response:**

Due to the steady-state assumption, the Gaussian plume model that was used to relate OCO-2 $XCO_2$ enhancements with emission sources is not reliable for long-range atmospheric transport (> 50 km, US EPA, 2015). We therefore prefer to restrict our analysis to the enhancements that can be related to sources within 50 km, thereby avoiding plumes originating from further away. For the cases that we selected, the agreement with the MEIC inventory (Fig. 3) suggests that we do not need to account for large emission sources outside the 50 km radius to interpret the enhancement.

- According to Bieser et al. (https://doi.org/10.1016/j.envpol.2011.04.030), roughly 90% of emissions from power plants occur between 200 m and 500 m above surface. How would emission estimates for power plants change using an average wind speed over this range rather than an average over 0–500 m (page 6, line 176)? Note that at the small distances between source and OCO-2 track considered in this study one cannot expect a homogeneous mixing of the plume over the depth of the PBL.

**Response:**

The 60 plumes that we analyzed are all from cities and industrial regions that have emission sources with various stack heights. The small industrial boilers and kilns, the major sources of $CO_2$ emissions in China, typically have smokestacks that are several tens of meters high. Therefore we used an average wind over 0–500 m to estimate the cross-sectional $CO_2$ fluxes from cities and industrial regions, which is the same configuration as Beirle et al. (2011) who also estimated city emissions (of nitrogen oxide) based on satellite observations.

Minor points and grammar:

- Page 1, line 18: Change "from the detailed China's emission inventory" to "from China's detailed emission inventory"

**Response:**

Corrected.

- P2, L34: "with the footprints" –> "with footprints"

**Response:**

Corrected.

- P2, L35: "natural $CO_2$ budget" -> "natural $CO_2$ budgets"

**Response:**

Corrected.

- P2, L36: "has allowed the initial insight" -> "has provided initial insight"

**Response:**

Corrected.

- P2, L46: "spaceborne $CO_2$ observation" -> "spaceborne $CO_2$ observations"

**Response:**

Corrected.

- P3, L64: "relies on the information about the wind" -> "relies on auxiliary information about winds"

**Response:**

Corrected.

- P3, L66: "provides the location" -> "provides the locations"

**Response:**

Corrected.

- P3, L75: "satellite sampling of OCO-2 capability" -> "sampling capability of OCO-2"

**Response:**

Corrected.

- P3, L77: "centered at the locations" -> "centered on the locations"

**Response:**

Corrected.

- P3, L86: Why should several $XCO_2$ anomalies belong to the same $CO_2$ plume? There is only a single transect per plume. Because of the moving windows?

**Response:**

The $XCO_2$ anomalies are those exceeding two sigmas of the spatial variability above the local average in each moving window. If a $CO_2$ plume crosses an OCO-2 track, the OCO-2 should observe a plume transect with $XCO_2$ enhancement, where there could be several $XCO_2$ anomalies larger than two sigmas above the local mean, although they correspond to the same $CO_2$ plume.

- P4, L91: "8 footprints if no is missing" -> "8 footprints if none is missing"

**Response:**

Corrected.

- P4, L93: "within $CO_2$ plume" -> "within the plume"

**Response:**

Corrected.

- P4, L104: Only a detail: Why is the along-wind distance measured in kilometres, but the across-wind distance in meters?

**Response:**

Here $z$ (along-wind distance) has to be specified in kilometers to give $a \cdot z^{0.894}$ in meters (Bovensmann et al., 2010).

- P5, L136: I think it would be clearer to state "We find 49,322 cases with local $XCO_2$ enhancements". It wasn't clear to me initially whether these were individual pixels or plumes.

**Response:**

Corrected.

- P5, L144: 50 km is not an appropriate scale for synoptic transport. I suggest to simplify to ".. or transport of $CO_2$ over a longer distance"

**Response:**

Corrected.

- P5, L148: "in space to make it difficult" -> "in space making it difficult"

**Response:**

Corrected.

- P6, L153: It would be better to write "Although the total number of selected cases is small, it is several times larger .."

**Response:**

Corrected.

- P6, L163: "at about local 13:30" -> "at about 13:30 local time"

**Response:**

Corrected.

- P6, L164: "part of OCO-2 ground track" -> "part of the OCO-2 ground track"

**Response:**

Corrected.

- P6, L175: "$CO_2$ fluxes" -> "$CO_2$ flux"

**Response:**

Corrected.

- P6, L178: Why shifted by 1°? Maybe it would be clearer to state "shifted by 1° in this case".

**Response:**

Corrected.

- P7, L186: How was the uncertainty of the hourly emission rate of Qinhuangdao determined? Does the MEIC inventory include uncertainties?

**Response:**

Yes, the MEIC inventory includes uncertainties of city emission estimates (Zheng et al., 2018).

- P7, L194: There were 4 cases where the same source was quantified twice. It would be good to know how consistent those double quantifications are with the estimated uncertainty of <24%.

**Response:**

The emissions from 3 cities were quantified twice over the same season (i.e., cold or warm) at the same or different years. The consistency in these estimates for the same city, defined as the difference between one estimate and the two independent estimates mean, is 15–24%.

- P7, L203ff: The interpretation of the small differences of 5-6% between satellite based estimates and MEIC in different seasons is pushed too far in this section considering the uncertainties. At least the arguments should be presented as possible explanations rather than as facts (e.g. write "could be due to" rather than "are due to"). The over-interpretation of the results culminates in the statement that human respiration accounts for 38% of the (5.5%) difference and that the remaining difference could be due to a bias in MEIC. The numbers deduced from the satellite observations are not sufficiently robust to speculate about a bias in the inventory as small as 3 percent. Uncertainties in the method (notably the assumption that the 0-500 m average wind speed is representative) could easily explain such differences, probably also differences in the results between summer and winter since vertical mixing is different in these seasons.

**Response:**

We rewrote this paragraph as follows according to the reviewer's suggestion.

"The differences in the results between cold and warm seasons could be due to uncertainties in the emission estimate methods of both our OCO-2 based inversion and the MEIC inventory. The satellite-based larger estimates in the cold season could be partially due to the fact that human respiration contributes to urban $CO_2$ fluxes while not included in the MEIC inventory of fossil fuel and cement emissions. We make a rough estimate of the metabolic $CO_2$ release by multiplying an emission factor of 0.52 t-$CO_2$ yr$^{-1}$ person$^{-1}$ (Prairie and Duarte, 2007) by the population living in each emitting area. The results suggest that human metabolic $CO_2$ emissions explain 8% of the larger satellite-based emission estimates on average in the cold season. The remaining difference could be due to the assumption that the 0-500 m average wind speed is representative of the transport wind in the plume diffusion, the natural processes like plant respiration, or the slight growth of fossil fuel emissions since 2013, but could also reflect some bias in the MEIC estimates. In the warm season, despite human respiration emissions, the satellite-based inversions give lower emission estimates possibly due to the carbon uptake by plants damping the $XCO_2$ enhancements (Mitchell et al., 2018), which makes anthropogenic emission signals not easily separated from the background in the satellite-based inversions."

- P8, L233: "principle, not all" -> "principle, but not all"

**Response:**

Corrected.

- P8, L235: "densely urban areas" -> "densely populated urban areas"

**Response:**

Corrected.

- P8, L44: The last section should be renamed to "Conclusions".

**Response:**

Corrected.

- P9, L271: "with less $CO_2$ inventory infrastructures" -> "with less advanced $CO_2$ inventory infrastructures"

**Response:**

Corrected.

- Figure 1: The blue bar should be called "$XCO_2$ anomalies" rather than "$XCO_2$ outliers"

**Response:**

Corrected.

**References**

Beirle, S., Boersma, K. F., Platt, U., Lawrence, M. G., and Wagner, T.: Megacity Emissions and Lifetimes of Nitrogen Oxides Probed from Space, Science, 333, 1737-1739, doi: 10.1126/science.1207824, 2011.

Bovensmann, H., Buchwitz, M., Burrows, J. P., Reuter, M., Krings, T., Gerilowski, K., Schneising, O., Heymann, J., Tretner, A., and Erzinger, J.: A remote sensing technique for global monitoring of power plant $CO_2$ emissions from space and related applications, Atmos. Meas. Tech., 3, 781-811, doi: 10.5194/amt-3-781-2010, 2010.

US EPA: Revision to the Guideline on Air Quality Models: Enhancements to the AERMOD Dispersion Modeling System and Incorporation of Approaches to Address Ozone and Fine Particulate Matter. Tech. rep. US Environmental Protection Agency #2060-AS54. https://www3.epa.gov/ttn/scram/11thmodconf/9930-11-OAR_AppendixW_Proposal.pdf, 2015.

Zheng, B., Zhang, Q., Davis, S. J., Ciais, P., Hong, C., Li, M., Liu, F., Tong, D., Li, H., and He, K.: Infrastructure Shapes Differences in the Carbon Intensities of Chinese Cities, Environ. Sci. Technol., doi: 10.1021/acs.est.7b05654, 2018.